# SpaceNet: A Multimodal Fusion Architecture for Sound Source Localization in Disaster Response

**DOI:** 10.3390/s26010168

**Published:** 2025-12-26

**Authors:** Long Nguyen-Vu, Jonghoon Lee

**Affiliations:** AI Laboratory, MoAdata, Seongnam-si 13449, Gyeonggi-do, Republic of Korea; longnv@moadata.ai

**Keywords:** sound source localization, TDoA, neural network, digital signal processing, feature fusion

## Abstract

Sound source localization (SSL) has evolved from traditional signal-processing methods to sophisticated deep-learning architectures. However, applying these to distributed microphone arrays in adverse environments is complicated by high reverberation and potential sensor asynchrony, which can corrupt crucial Time-Difference-of-Arrival (TDoA) information. We introduce SpaceNet, a multimodal deep-learning architecture designed to address such issues by explicitly fusing audio features with sensor geometry. SpaceNet features: (1) a dual-branch architecture with specialized spatial processing that decomposes microphone geometry into distances, azimuths, and elevations; and (2) a feature-normalization technique to ensure stable multimodal training. Evaluation on real-world datasets from disaster sites demonstrates that SpaceNet, when trained on ILD-only mel-spectra, achieves better accuracy compared to our baseline model (CHAWA) and identical models trained on full mel-spectrograms. This approach also reduces computational overhead by a factor of 24. Our findings suggest that for distributed arrays in adverse environments, time-invariant ILD cues are a more effective and efficient feature for localization than complex temporal features corrupted by reverberation and synchronization errors.

## 1. Introduction

Sound Source Localization (SSL) is the fundamental signal-processing and machine-learning task of estimating the position of one or more active sound sources from signals captured by an array of spatially distributed microphones. The position is typically represented as a Direction of Arrival (DoA), comprising azimuth and elevation angles, or as a full set of three-dimensional Cartesian coordinates relative to a reference point, often the center of the microphone array. This capability is a cornerstone of computational auditory scene analysis, enabling a wide range of applications from human-robot interaction and autonomous navigation to smart home assistants, augmented reality, and acoustic surveillance [1,2].

The field of SSL is closely related to, yet distinct from, several other audio-processing tasks such as Sound Event Detection (SED) which focuses on identifying what sounds are present in an audio stream and when they occur, without determining their spatial origin. The combination of these two tasks gives rise to a more complex and comprehensive problem known as Sound Event Localization and Detection (SELD), which aims to simultaneously estimate the class, temporal activity, and DoA of all active sound sources in a scene, providing a rich spatio-temporal understanding of the acoustic environment. The advancements in SELD are intrinsically linked to progress in SSL, as robust localization is a prerequisite for accurate SELD performance [3].

For decades, SSL has been dominated by classical model-based signal-processing techniques. These methods rely on well-defined physical models of sound propagation. Prominent examples include approaches based on the Time Difference of Arrival (TDoA), which estimate the relative delays of a sound signal reaching different microphone pairs to triangulate the source position. Another foundational technique is the Steered Response Power (SRP) method, often augmented with a Phase Transform (PHAT) weighting (SRP-PHAT), which systematically scans a grid of potential source locations and computes the acoustic power, identifying the location with the maximum response as the source position. High-resolution spectral estimation algorithms, such as Multiple Signal Classification (MUSIC), offer another powerful approach by exploiting the subspace properties of the received signals’ covariance matrix.

Although these classical methods are mathematically elegant and effective in anechoic or moderately reverberant conditions, their performance is known to degrade substantially in the complex and adverse acoustic scenarios commonly encountered in the real world. The presence of significant background noise, strong and prolonged reverberation, and the simultaneous activity of multiple sound sources can severely distort the spatial cues upon which these algorithms depend, leading to inaccurate or ambiguous localization estimates. This performance gap between controlled and real-world environments has been the primary catalyst for a paradigm shift in the field.

The limitations of classical methods have driven the research community to embrace data-driven approaches, leading to a revolution in SSL powered by deep learning. Deep Neural Networks (DNNs) have demonstrated the ability to learn robust and discriminative features directly from acoustic data, which prove to be more resilient to noise and reverberation than their model-based counterparts. In this new paradigm, the SSL task is typically framed as either a regression problem, where the network directly outputs the source coordinates or angles, or a classification problem, where the network predicts a probability distribution over a discrete grid of possible locations [4,5].

Despite this progress, current deep-learning-based SSL methods may exhibit fundamental shortcomings if they are not limited to the setting of a compact and fixed microphone array. Most architectures treat localization as a purely acoustic problem, overlooking the fact that microphone array geometry encodes critical spatial constraints that govern sound propagation [3,6]. Attempts to incorporate spatial cues often reduce them to handcrafted cross-channel features such as GCC-PHAT, which are sensitive to noise and reverberation [7,8,9]. Furthermore, when spatial information is integrated explicitly, possible mismatches of spectral audio features and bounded geometric parameters may lead to training instability [2]. This lack of principled multimodal fusion prevents existing systems from achieving the robustness required for demanding real-world applications such as search and rescue (SAR) [10,11], where microphone geometry and acoustic scene complexity cannot be ignored. These limitations highlight the need for architectures that explicitly unify audio and spatial modalities within a physics-informed design.

In this study, we propose SpaceNet, a multimodal neural network architecture that utilizes both acoustic features and spatial features explicitly to improve the performance of sound source localization in terms of accuracy and coefficient of determination. The network branch that processes audio features is derived from ResNet18 [12], whereas the branch that encodes spatial features retrieved from microphones’ positions is handcrafted with trainable weights and careful fusion. We address the incompatibility issues of the two network branches with z-score normalization so that the fully connected layer can be easily trained. SpaceNet exhibits significant improvement over our previous work—CHAWA [13]—in challenging, realistic environments in search-and-rescue (SAR) operations. Similar to our previous work, we aim to improve the performance of a sound source localization system using a distributed microphone array, evaluated on different challenging datasets retrieved from realistic SAR scenarios such as ruined buildings.

In the next sections, we go through the background and related work in Section 2. We then describe the design philosophy and architecture of SpaceNet in Section 3. The dataset experimental analysis is conducted in Section 4 and Section 5. Finally, we conclude our work in Section 6.

## 2. Background

### 2.1. SSL Applications in Disaster Response

The use of sound source localization (SSL) in disaster management and search-and-rescue (SAR) operations has received increasing attention in recent years [10]. Early studies demonstrated the feasibility of integrating auditory systems into rescue platforms. For example, ref. [14] developed an SSL-based auditory module to enable robots to detect survivors’ cries for help, while ref. [11] proposed the use of bio-inspired “biobots,” where Madagascar hissing cockroaches were equipped with miniature microphone arrays to localize victims’ sounds.

Other approaches have investigated aerial and mobile platforms equipped with microphone arrays. Microphone systems mounted on unmanned aerial vehicles (UAVs), helicopters, and drones have been applied to detect and estimate the location of human sound sources in disaster zones [15,16,17]. These studies highlight the potential of SSL to support rapid victim detection when visual cues are unavailable due to debris, darkness, or adverse conditions.

A persistent challenge across these works is robustness to environmental noise. Disaster scenes are often dominated by strong ambient sounds such as wind, rain, flowing water, heavy machinery, and vehicle activity, which significantly degrade localization performance. Addressing these acoustic challenges remains a key requirement for deploying SSL systems in real-world SAR scenarios.

### 2.2. Deep-Learning Approaches Exploiting Spatial Information in SSL

Deep learning has reshaped sound source localization (SSL), offering robustness to noise and reverberation beyond classical model-based methods. Early approaches typically applied deep neural networks (DNNs) to spectral or cross-channel features such as GCC-PHAT, TDoAs, or interaural level differences (ILDs), and framed SSL as either classification over discrete angular bins or regression to continuous coordinates [4,5,18,19,20,21,22,23]. CNNs proved effective at extracting spatially structured representations, while RNNs captured temporal context, and CRNN hybrids were employed for joint detection, localization, and tracking [21,22]. Variants such as T–F masking combined with GCC-PHAT or steering vectors further improved direction-of-arrival (DoA) estimation in reverberant conditions by enhancing phase-dominant time–frequency bins [24].

Subsequent studies expanded input features to better capture spatial cues. Works exploited eigenvectors of the spatial coherence matrix, SRP-PHAT power maps, and causal 3D CNNs for multi-source tracking under strong reverberation [25]. These methods underscored the benefit of explicitly leveraging inter-channel phase and spatial correlation, though robustness across arrays remained limited.

A growing body of research has emphasized explicit incorporation of array geometry and multimodal spatial features. wav2pos [26] introduced a masked autoencoder combined with NGCC-PHAT and SincNet backbones, fusing microphone position information with audio embeddings to achieve geometry-aware generalization. Recent architectures such as TF-Mamba [27] used bidirectional time–frequency fusion with spatial cues for resilience in noise, while SoundLoc3D [1] extended SSL to multimodal RGB-D acoustic cameras for 3D localization and classification. Other works further highlight this trend: visual–acoustic co-localization [28], object-aware scene understanding [29], efficient microphone-fault-tolerant 3D SSL [30], and hybrid CRNNs tailored for undesirable conditions [2]. These efforts reinforce the centrality of geometry and multimodal integration in advancing SSL performance.

Nevertheless, challenges persist. Many existing models depend on handcrafted spatial preprocessing (e.g., GCC-PHAT) sensitive to noise, or they require additional sensing modalities not feasible in SAR and other real-world deployments. Moreover, mismatched scales between acoustic features and bounded spatial parameters often cause training instabilities. These limitations motivate the design of unified frameworks, such as our proposed SpaceNet, which fuses audio and spatial features through normalization and principled multi-branch integration to achieve robust SSL in realistic and dynamic environments.

## 3. SpaceNet

### 3.1. The SpaceNet Paradigm

SpaceNet represents a shift in SSL architecture design, moving beyond conventional approaches that treat localization as a pure audio-processing problem. The nomenclature of SpaceNet implies the Spatial information and the use of ResNet18 backbone. Our study introduces a principled framework for multimodal SSL that addresses each of the identified limitations, taking into account that the geometric positions of the microphone sensors can dynamically change. The architecture is built on three core innovations:

#### 3.1.1. Explicit Spatial Decomposition

SpaceNet introduces a neural architecture that explicitly decomposes spatial microphone geometry into its constituent components (distances, azimuths, elevations) and processes each component through separated pathways with learnable aggregation mechanisms.

#### 3.1.2. Principled Feature Normalization

We develop a normalization technique that addresses the fundamental challenge of multimodal training instability through modality-specific statistical normalization, enabling stable convergence and optimal feature fusion.

#### 3.1.3. Physics-Informed Fusion

Our fusion strategy incorporates domain knowledge about acoustic propagation through learnable weighting mechanisms that allow the network to adaptively balance audio and spatial information based on their informativeness for localization.

### 3.2. Architectural Design Principles

The SpaceNet architecture shown in Figure 1 is built upon the principle that effective sound source localization (SSL) requires explicit modeling of the relationship between acoustic propagation and array geometry. Unlike approaches that rely on implicit encoding through multi-channel audio features, SpaceNet explicitly separates and processes spatial information through a dedicated pathway.

Modality Specialization: Each modality (audio and spatial) requires tailored processing that respects its unique structure. Audio features encode temporal-spectral patterns that benefit from hierarchical convolutional processing, while spatial features represent geometric relationships that require distinct mathematical transformations.Explicit Spatial Decomposition: Microphone geometry contains heterogeneous types of information (distances, azimuth angles, elevation angles) that should be extracted and processed separately before fusion. This decomposition enables the network to learn specialized representations for different geometric aspects.Learnable Fusion Weights: The relative contribution of modalities and spatial components varies across environments and should be adaptively learned, rather than predetermined.

The decomposition of SpaceNet deep layers and their corresponding learnable weights can be found in Table 1. Next, we dive deeper into the design of SpaceNet architecture to further justify these principles.

#### 3.2.1. Audio-Processing Branch

The audio branch leverages a ResNet18 backbone adapted for multi-channel acoustic inputs with the two properties: (1) Multi-channel Spatial Awareness: Standard ResNet applications process three-channel RGB images, but here the first convolutional layer is modified to accept four-channel spectrograms. This preserves microphone-wise spatial information and allows the network to learn cross-channel correlations that encode acoustic spatial cues; (2) Hierarchical Feature Learning: ResNet18 processes multi-channel spectrograms through four residual stages, producing increasingly abstract representations. Early layers capture local time–frequency patterns, while deeper layers integrate broader temporal and cross-channel structures that are critical for localization.

#### 3.2.2. Spatial Processing Branch

The spatial branch introduces SpaceNet’s innovation compared to CHAWA [13], which is the explicit modeling of microphone geometry through decomposed spatial features. The spatial encoding framework captures the relative arrangement of microphones, which varies across recordings. Direct use of raw (x,y,z) coordinates is straightforward but fails to highlight the specific geometric cues that influence localization. Instead, we encode features based on centroid-relative distances, azimuths, and elevations—quantities widely used in SSL. The centroid acts as an anchor for deriving these features, as illustrated in Figure 2.

Geometric Feature Extraction: Given four microphone coordinates ${\left\{m_{i}\right\}}_{i=1}^{4}$, spatial features are extracted by: (1) computing the centroid for translation invariance, (2) deriving relative positions that preserve array geometry, (3) converting to spherical coordinates (distance, azimuth, elevation) that isolate distinct geometric components.

Component-wise Processing: The resulting 12-dimensional spatial feature vector is decomposed into three groups:Distances $d\in R^{4}$: encode array size and microphone spacing,Azimuths $\alpha \in R^{4}$: capture horizontal angular structure,Elevations $\beta \in R^{4}$: capture vertical configuration.

Each group is passed through a dedicated fully connected transformation:$$\begin{array}{cc} h_{d} & =\tanh(W_{d}d+b_{d})\\ h_{\alpha } & =\tanh(W_{\alpha }\alpha +b_{\alpha })\\ h_{\beta } & =\tanh(W_{\beta }\beta +b_{\beta }) \end{array}$$
where $W_{d}$, $W_{\alpha }$, and $W_{\beta }$ are learnable parameters for each component, and $b_{d}$, $b_{\alpha }$, and $b_{\beta }$ are biases. This separation ensures specialized learning for each spatial component.

Learnable Aggregation: Processed components are aggregated into scalar descriptors using learnable weights:$$z_{i}=w_{i}\cdot \tanh\left(w_{agg}^{T}h_{i}+b_{agg}\right),$$
where i∈{d,α,β}. This mechanism allows adaptive emphasis on different spatial components depending on the environment and source position.

We next introduce the detailed spatial encoding method used in SpaceNet.

### 3.3. Spatial Geometry Decomposition

#### 3.3.1. Centroid-Based Coordinate System

For each audio sample i=1,…,N, let the microphone positions be ${\left\{m_{l}\right\}}_{l=1}^{4}$. Compute the array centroid:$$c=\frac{1}{4}\sum_{l=1}^{4}m_{l}$$

Then compute relative vectors from the centroid:$$r_{l}=m_{l}-c,\;l=1,\ldots ,4$$

This re-centers the coordinate system at the array centroid, providing translation invariance while preserving array geometry.

#### 3.3.2. Spherical Decomposition

Relative vectors are mapped to spherical coordinates to extract distinct geometric components:$$\begin{array}{cccc} d_{l} & =∥r_{l}{∥}_{2} & & (\mathrm{radial}\;\mathrm{distance})\\ \alpha_{l} & =\arctan2(r_{l,y},r_{l,x}) & & (\mathrm{azimuth}\;\mathrm{angle})\\ \beta_{l} & =\arctan2(r_{l,z},\sqrt{r_{l,x}^{2}+r_{l,y}^{2}}) & & (\mathrm{elevation}\;\mathrm{angle}) \end{array}$$

The 12-dimensional spatial feature vector for sample *i* is then:$$x_{i}^{\left(s\right)}={\left[d,\alpha ,\beta \right]}^{T}\in R^{12}$$

### 3.4. Normalization

Given a dataset $D={\left\{\left(x_{i}^{\left(a\right)},x_{i}^{\left(s\right)},y_{i}\right)\right\}}_{i=1}^{N}$, we normalize both the audio features $x_{i}^{\left(a\right)}$ and spatial features $x_{i}^{\left(s\right)}$ to ensure effective multimodal fusion and stable gradient flow during training given the target coordinates $y_{i}$ (speaker positions).

#### 3.4.1. Audio Feature Normalization

Global z-score normalization is applied across all audio features:$$\mu_{a}=\frac{1}{N\cdot d_{a}}\sum_{i=1}^{N}\sum_{j=1}^{d_{a}}x_{i,j}^{\left(a\right)}$$$$\sigma_{a}^{2}=\frac{1}{N\cdot d_{a}}\sum_{i=1}^{N}\sum_{j=1}^{d_{a}}{\left(x_{i,j}^{\left(a\right)}-\mu_{a}\right)}^{2}$$x^i(a)=xi(a)−μamax(σa,ϵ)
where $\epsilon =10^{-8}$ prevents division by zero.

#### 3.4.2. Spatial Feature Normalization

The 12-dimensional spatial feature vector $x_{i}^{\left(s\right)}$, derived as described in the Spatial Geometry Decomposition section, is normalized using a per-dimension z-score.

For each dimension k=1,…,12 of the spatial feature vector, we compute its mean and standard deviation across the entire dataset:$$\mu_{s,k}=\frac{1}{N}\sum_{i=1}^{N}x_{i,k}^{\left(s\right)}$$$$\sigma_{s,k}^{2}=\frac{1}{N}\sum_{i=1}^{N}{\left(x_{i,k}^{\left(s\right)}-\mu_{s,k}\right)}^{2}$$

The normalized feature x^i,k(s) is then calculated as:x^i,k(s)=xi,k(s)−μs,kmax(σs,k,ϵ)

## 4. Datasets

### 4.1. Naming Convention

For convenience, we named a dataset based on the location where its samples were retrieved. For instance, we have recorded in the ruined buildings of Gyeonggi Fire Academy, pumping water stations in Yeongdong (underground) and Yeoju (basement), and a collapsed building in Goyang, as shown in Figure 3.

### 4.2. Components

The components were guided by several factors and lessons learned from the previous work [4,5,13], for instance:Microphones: The arrangement and number of microphones are dynamic, i.e., microphones are placed arbitrarily prior to recording a sample. We improved the hardware constraints in our previous work using wireless microphones in addition to wired ones. The number of microphones was 4, posing challenges to localization models. The brands of our devices include 4 RØDE Wireless GO II microphones (RØDE Microphones, Sydney, Australia) with 2 receivers and a Zoom R24 mixer (Zoom Corporation, Tokyo, Japan) to receive data from the receivers before sending it to the computer.Geometry: Our experiments include both 2D and 3D settings. In fact, due to the complexity and high-risk nature of the work of recording, we refrained from setting up variable depths of the microphones but focused more on diversifying the width and height instead. Therefore, we expect that 2D experiments can be more realistically reflected. In either case, spherical coordinates (azimuth, elevation, and distance) or Cartesian coordinates can be applied, given a transformation of the z-axis and elevation angle.Sound Source: We assume the victim (sound source) is stationary, thus the loudspeaker playing the audio samples from Figure 4 remained fixed for one 4-channel audio sample, and then relocated for the next recording.Environment: All datasets introduced realistic ambient noise artifacts such as machine running, helicopter flying nearby, and echoes from the wall.

This design deliberately reflects the challenges of sound source localization in realistic conditions. The quadrilateral microphone geometry amplifies the difficulty of time-difference-of-arrival (TDoA) estimation, while unpredictable ambient noise and reverberation at the sites and synchronization across independent USB/wireless microphones introduce additional uncertainties. Such imperfections make the datasets challenging testbeds for evaluating machine-learning approaches to localization, where robustness to synchronization errors and robustness to noise are essential.

### 4.3. Environment Setup

In our previous work [13], the SoS dataset was recorded in a large anechoic convention hall of size 26.5×15.5 m^2^. In our Yeoju dataset, we used a larger space with dimension 17.9×29×7.06 m^3^. In the Gyeonggi Fire Academy dataset, space is more compact due to the limitation of the ruined building size of 9.25×5.05×5.05 m^3^. For the Yeongdong dataset, the dimensions are 15.1×6.1×8.5 m^3^. Meanwhile, the collapsed building in Goyang and its surrounding area have size of 8.0×8.0×5.0 m^3^ approximately. Among these, we used wireless microphones for the Yeoju dataset due to the limitation of cable length. For more detail in the setup, in the Yeoju dataset, we moved the loudspeaker between 98 locations while varying the four microphones’ positions as well (4 settings of the distributed microphone array).

This time, we did not need to simulate impulse response and noise, but only needed to utilize real-life recorded audio samples of people/animals in distress to be played at the loudspeaker (sound source) as depicted in Figure 4. The microphones and the loudspeaker were dynamically placed at different positions during the recording session. Four microphones simultaneously recorded the audio samples generated by the loudspeaker, resulting in 4-channel audio data.

### 4.4. Dataset Description

The datasets were collected using distributed microphone arrays, where four microphones were positioned at the corners of each recording site. For every session, approximately 30 s of audio was captured simultaneously across all four channels. These recordings were segmented into clips of 3.0 s with an overlap of 2.5 s. Since each set of four channels corresponds to a single recording session, the effective number of independent samples is one quarter of the raw segment count. After this adjustment, the Yeoju dataset contains 14,708 samples. The Yeongdong dataset contains 1056 samples. The Fire Academy dataset provides 3342 samples. Metadata for each segment includes the four audio tracks, speaker identity, and Cartesian coordinates of the sensors and the sound source (speaker) location. A summary of the statistics of the datasets, except for the Goyang dataset, is presented in Table 2.

## 5. Experiments

This section evaluates our two central claims: (1) SpaceNet achieves improved localization performance compared to CHAWA [13]; and (2) the proposed normalization and spatial information modules each contribute positively to the performance of SpaceNet, as shown in an ablation study.

### 5.1. Evaluation Methodology

#### 5.1.1. Datasets and Dimensionality

Due to the scale of audio collection and the workload required for cleaning and preprocessing, we selected three datasets: the Gyeonggi Fire Academy, Yeongdong, and the Yeoju Water Pumping Station for model comparison, and Yeoju as a representative dataset for the ablation study. Disaster environments posed constraints on data collection, particularly along the depth axis, since placing loudspeakers at multiple vertical levels was impractical and unsafe. Consequently, the variance along the *z*-axis was limited, which reduces statistical robustness in full 3D scenarios. To mitigate this, we conducted some experiments in the ablation study section in both 2D and 3D spaces, ensuring a realistic yet tractable evaluation setting. In 2D settings, we actually did not ignore the *z*-axis (elevation) but rather considered *x* and *y* axes only for calculating the distance between the ground truth and predicted locations. For three-second segments, mel-spectrogram has a size of 512 × 259, mel-spectra has a size of 4 × 128. During training, the input to the neural networks would have the shape of [*B* × *C* × *W* × *H*] where *W* × *H* can be either 512 × 259 or 4 × 128. *B* is the batch size, which is defaulted to 128 in our experiments. The channel *C* is always 1 since this task does not involve RGB images.

#### 5.1.2. Coordinate Representation

Cartesian representation avoids angular wrap-around discontinuities (e.g., at $-180^{\circ }/180^{\circ }$), thereby enabling smoother regression learning in neural networks [19,21]. This choice also simplified the recording process, eliminating the need for separate azimuth and elevation measurements. While classical methods such as SRP-PHAT rely on spherical coordinates due to their grid-based search over azimuth and elevation [23,25], Cartesian and spherical formulations remain mathematically equivalent, and the optimal choice depends on the algorithm and microphone array geometry. In our work on disaster response, we chose Cartesian coordinates to localize the sound source emitted by the victims, as demonstrated in Figure 5.

#### 5.1.3. Comparison Scope

Our dataset does not include a “start signal from the ground truth system,” which made it incompatible with the LuViRA dataset and the wav2pos method [26]. In particular, wav2pos reported results primarily on synthetic data generated with Pyroomacoustics [31], limiting the relevance of direct comparison in our realistic setting.

#### 5.1.4. Experimental Framework

We implemented all experiments in PyTorch (version 2.4.1) [32]. Data were loaded using the PyTorch DataLoader and integrated with scikit-learn (version 1.7.0) to streamline the end-to-end pipeline and improve reproducibility. Model classification heads were reconfigured into multi-output regression heads. Training was performed on two Tesla T4 GPUs (16 GB NVRAM each) with an Intel Xeon Silver 4210 CPU @ 2.20 GHz.

#### 5.1.5. Features and Augmentation

In our prior work [13], we applied augmentation techniques such as time mask and frequency mask, along with magnitude-based features, to mitigate phase inconsistencies. In this study, we utilized features commonly used in the DCASE challenge [3], specifically mel-spectra and log mel-spectrograms, which better capture frequency content on the mel scale for localization tasks. We note that frequency masking and time masking [33,34] can be used to test complementary effects at both signal and spectral levels. However, for some experiments, such as the model comparison and ablation studies, we avoided using augmentation to observe the raw potential of each model-feature setting. In particular, this ensured that improvements in SpaceNet’s performance could be attributed directly to the normalization and spatial information modules rather than augmentation effects. Last but not least, all training runs employed the Huber loss and the AdamW optimizer by default.

#### 5.1.6. Training and Inference

Training was performed using a batch size of 128, with the AdamW optimizer and Huber loss function. Inference was conducted in a batched manner, with the model outputting predicted Cartesian coordinates for each input sample. The initialization phase to calculate the mean and standard deviation for normalization took approximately 72 s, given 5427 samples, assuming that each sample has a length of 3 s. The training and testing times varied based on the dataset size, but take 218 s per epoch.

### 5.2. Model Comparison

This section discusses the performance of SpaceNet versus CHAWA. We evaluated the two models against three datasets: Yeongdong, Yeoju, and Gyeonggi without using any augmentation techniques. As shown in Table 3, SpaceNet leads the benchmarking on both the Yeongdong and Yeoju datasets, performing better than all model instances of CHAWA by a large margin, regardless of the features used, across nearly all radius thresholds. For the Gyeonggi dataset, SpaceNet was able to keep pace with CHAWA with a smaller margin. For instance, regarding the mel-spectra feature at the 2-m radius, CHAWA (67.5%) performs better than SpaceNet (66.5%) by only 1%.

To clarify, it is important to note that the Yeoju and Yeongdong sites represent more extreme acoustic environments: the former is a basement characterized by excessive echo (reverberation), while the latter is located underground in a water pumping station with substantial background noise. Consistent with our previous study [13], CHAWA appears to be more robust in smaller spaces. Specifically, the Yeoju dataset was collected in a space of 3664 m^3^, the Yeongdong dataset in 782 m^3^, and the Gyeonggi dataset in only 235 m^3^. The narrower spatial dimensions may have played an important role in these results. It can also be observed that for the largest space (Yeoju), CHAWA’s performance declined significantly compared to its performance on the Yeongdong dataset. At the 2 m radius threshold, CHAWA achieved its best accuracies of 18.7% and 33.8% on the Yeoju and Yeongdong datasets, respectively, whereas they are 73.1% and 38.2% for SpaceNet.

In addition to localization accuracy, we also compared the two models in terms of Mean Absolute Error (MAE) and Root Mean Square Error (RMSE) under rainy and non-rainy conditions, as shown in Table 4. It can be observed that SpaceNet consistently outperforms CHAWA in both weather conditions. Notably, SpaceNet’s performance degrades only slightly in rainy conditions, with MAE increasing from 0.760 m to 0.900 m, and RMSE from 0.877 m to 0.926 m. In contrast, CHAWA’s performance shows a more pronounced degradation, with MAE decreasing from 1.962 m to 2.108 m, and RMSE from 1.970 m to 2.128 m. This indicates that SpaceNet is more robust to adverse weather conditions compared to CHAWA. The weather condition did not affect CHAWA; its performance was still worse than SpaceNet in both cases.

Next, we discuss the contribution of the audio-spatial encoding modules and normalization techniques to the robustness of the SpaceNet model based on the results of the ablation studies.

### 5.3. Ablation Studies

Systematic ablation studies confirmed the importance of each architectural component. We compare the performance of SpaceNet and its variants, from which the normalization and spatial modules were removed. The coefficient of determination ($R^{2}$) was used in addition to accuracy to observe how closely the predicted values match the actual data points (see example in Figure 6). As mentioned in Section 5.1, the ablation study was conducted in both 2D and 3D.

First, to test the impact of the spatial encoding module, we removed the spatial processing branch. In the 2D-localization task, this removal consistently resulted in performance degradation for both mel-spectra (74.2% to 72.9%) and mel-spectrograms (71.1% to 66.9%). However, we noticed inconsistencies in 3D-localization (Table 5, 3D, mel-spectrogram), where the ablated model performed competitively, or even slightly better, at some thresholds. This strongly suggests that the spatial module, as designed, is less effective when the *z*-axis variance in the dataset is already low, as noted in our methodology.

Second, to evaluate the effectiveness of the normalization technique, we found its impact was highly feature-dependent. For the mel-spectra feature, normalization proved critical: disabling it (comparing rows 2 and 3 of the 2D mel-spectra group) caused a significant accuracy degradation of 12.2% (from 65.5% to 53.3% at the 1.5-m radius). Conversely, for the mel-spectrogram feature (3D group), disabling normalization led to a slight increase in performance at the 1m-radius (32.6% vs. 35.0%). This suggests that the complex, full-temporal mel-spectrogram features have a distribution that our current normalization scheme struggles to scale effectively, whereas the simpler, time-averaged mel-spectra benefit significantly.

### 5.4. Spatial Compensation and Feature Regularization

Sound source localization traditionally relies on Inter-channel Level Difference (ILD) and Time Difference of Arrival (TDoA). We investigated the use of “mel-spectra” (This term was used interchangeably with mel-spectrogram in DCASE)—which we define as a mel-spectrogram averaged over the time dimension. This operation discards TDoA information and reduces the GPU workload by a factor of 6 for CHAWA and 24 for SpaceNet, leaving the model with only average ILD cues.

Counter-intuitively, this “degraded” feature produced promising results, particularly for our SpaceNet architecture. This is strongly evident in the model comparison (Table 3). On the Yeongdong and Yeoju datasets, SpaceNet equipped with mel-spectra significantly outperformed the same model using the full mel-spectrogram (e.g., at the 2-m radius threshold, 38.2% vs. 28.4% on Yeongdong, and 73.1% vs. 64.8% on Yeoju datasets). This finding is confirmed by our ablation study (Table 5), where mel-spectra also achieved higher accuracy in both 2D (74.2% vs. 71.1%) and 3D (73.1% vs. 64.8%) tasks.

This suggests that the full-temporal data in the mel-spectrogram may introduce redundant or noisy information, such as late reverberations, that can hinder localization. This can be explained by three following reasons: (1) The extreme reverberation in the Yeoju site (large basement), where the direct path is lost; and (2) The synchronization errors between the wireless microphones in the distributed array (max distance between microphones is about 30 m); (3) The time-averaging acts as an effective regularizer, forcing the model to learn from more stable, time-invariant ILD cues.

Given its performance and 24× computational savings, we identify mel-spectra as the optimal feature choice for this localization task.

### 5.5. Discussion

#### 5.5.1. Normalization Benefits

In addition to the experimental validation, we discuss two effects of normalization: gradient stability and feature scale compatibility.

Without normalization, the gradient magnitudes can vary significantly between the two neural network branches, given the different properties of audio and spatial information.$$\frac{\partial L}{\partial x^{\left(a\right)}}\propto ∥x^{\left(a\right)}∥,\;\frac{\partial L}{\partial x^{\left(s\right)}}\propto ∥x^{\left(s\right)}∥$$

Moreover, normalization ensures that both modalities contribute equally to the fusion.E[∥x^(a)∥]≈E[∥x^(s)∥]≈d
where *d* is the feature dimension. SpaceNet’s training dynamics benefit significantly from the proposed normalization framework, which addresses the fundamental challenge of multimodal training instability and exhibits the following convergence properties: (1) Reduced variance in gradient estimates; (2) Faster convergence to optimal solutions; and (3) Improved generalization performance.

#### 5.5.2. Microphone Geometry

It is important to note that the concept of equipping the main localization module with spatial features has been discussed in other studies, such as wav2pos [26]. In that work, the authors introduced microphone locations to an autoencoder model, then combined the Decoder’s output with the output of a pre-trained sub-network of GCC-PHAT with a SincNet backbone (NGCC-PPHAT). While both attempts make the architectures aware of the geometry settings of the microphones, wav2pos and SpaceNet are inherently different: First, SpaceNet uses a dedicated network branch to learn the microphone geometry, while wav2pos uses a pre-trained sub-network to learn TDoA features. Second, SpaceNet does not require pre-training and instead simultaneously trains the two network branches, thereby streamlining the training and validation process.

#### 5.5.3. Limitations and Future Work

This study has several limitations that open avenues for future work. Our experimental validation was conducted on three realistic, yet specific, disaster-site datasets. The practical and safety constraints of these environments limited the collection of audio with significant *z*-axis variance, restricting the full validation of 3D localization. Furthermore, while our ablation study revealed the conditional, feature-dependent benefit of our normalization scheme, its failure to improve performance on mel-spectrograms indicates that a more adaptive normalization strategy is required. For improvements, we plan to reconsider the design of the aggregation module, which collapses all distance information in SpaceNet, to better merge it with the audio branch. Throughout our experiments, the number of microphones was fixed at four; we decided not to lower this number due to the “cone of confusion” phenomenon, where multiple sound source locations yield nearly identical interaural time and level differences. In noisy environments, having fewer than four microphones could compromise localization accuracy. We, however, noticed that increasing the number of microphones, as in [26], could further improve the performance, which we plan to explore in future work.

## 6. Conclusions

In this paper, we introduced SpaceNet, a multimodal neural network architecture for sound source localization designed to aid SAR operations in disaster sites. We prepared multiple realistic datasets by recording audio samples from different locations, such as a water pump station and ruined buildings, to validate the performance of SpaceNet and its predecessor, CHAWA. By introducing an explicit spatial encoding module in addition to an audio-processing branch, and normalization techniques with learnable weights for compatible feature fusion, we believe that SpaceNet has shown progress in sound source localization for distributed microphone (sensor) settings. Facing the challenges of dataset collection and other technical difficulties, we recognize there is much room for improvement in SpaceNet, which we plan to address in future studies.

## Figures and Tables

**Figure 1 sensors-26-00168-f001:**
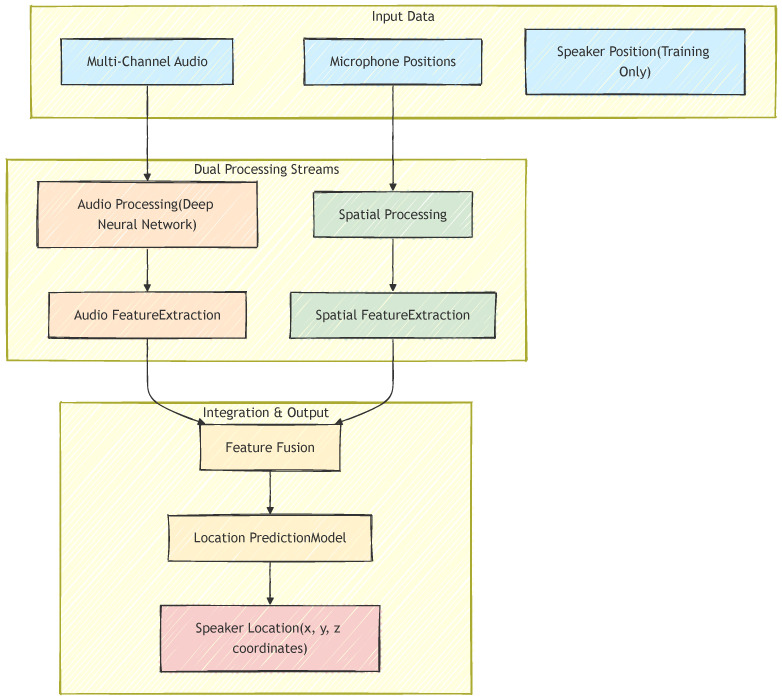
Overview of SpaceNet architecture: audio features and spatial features are fused for localizing the sound source.

**Figure 2 sensors-26-00168-f002:**
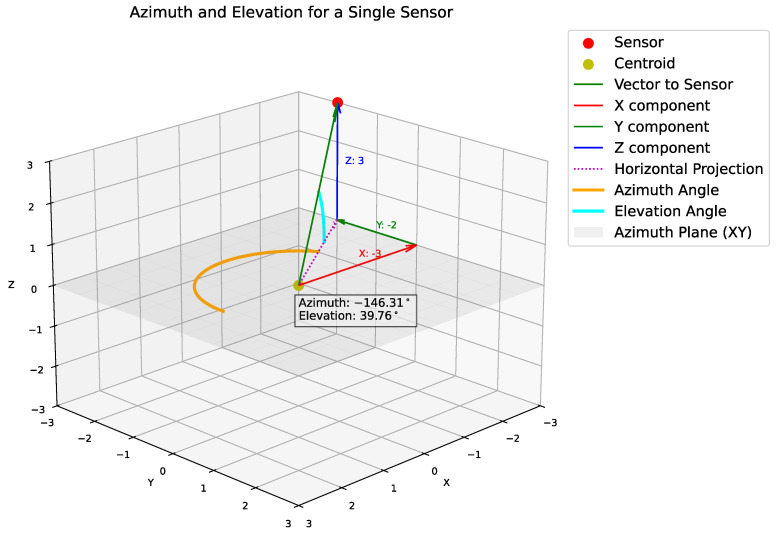
Centroid-to-sensor vectors that define the spatial geometry of the array.

**Figure 3 sensors-26-00168-f003:**
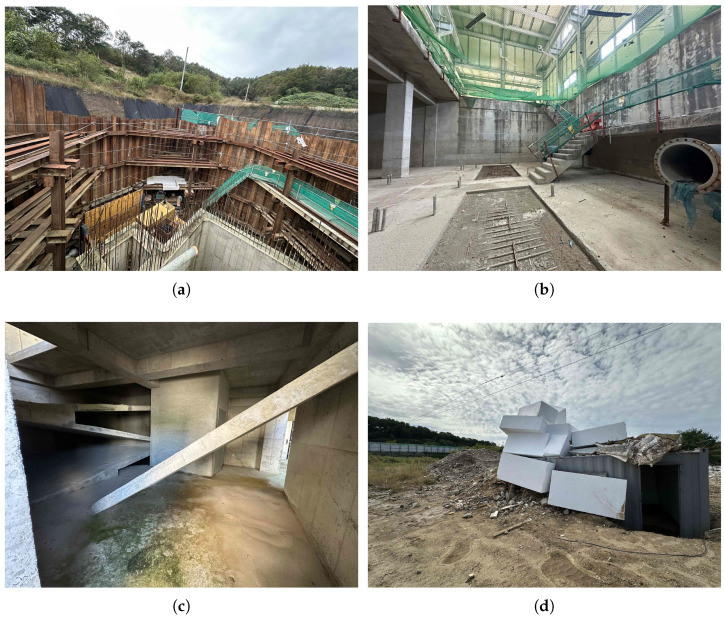
Replication of disaster sites where the audio samples were recorded. (**a**) Water Pumping Station, Yeongdong; (**b**) Water Pumping Station, Yeoju; (**c**) Fire Academy, Gyeonggi; (**d**) Collapsed Building, Goyang.

**Figure 4 sensors-26-00168-f004:**
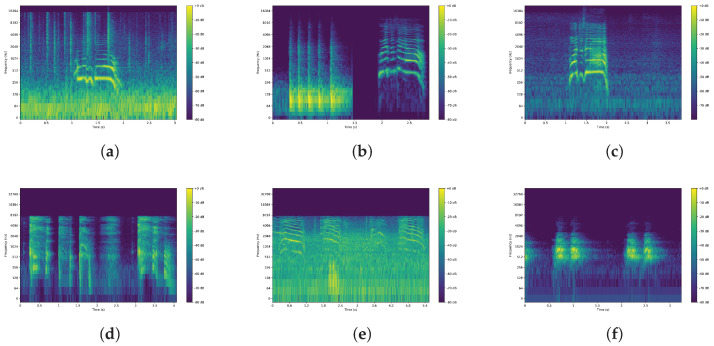
Spectrograms of sound samples to be played from the loudspeaker (sound source). We reused the samples from [13]. (**a**) Fire burning, Male voice-1 in distant; (**b**) Door pounding, Female voice; (**c**) Male voice-2; (**d**) Person coughing; (**e**) Cat yowling; (**f**) Dog barking.

**Figure 5 sensors-26-00168-f005:**
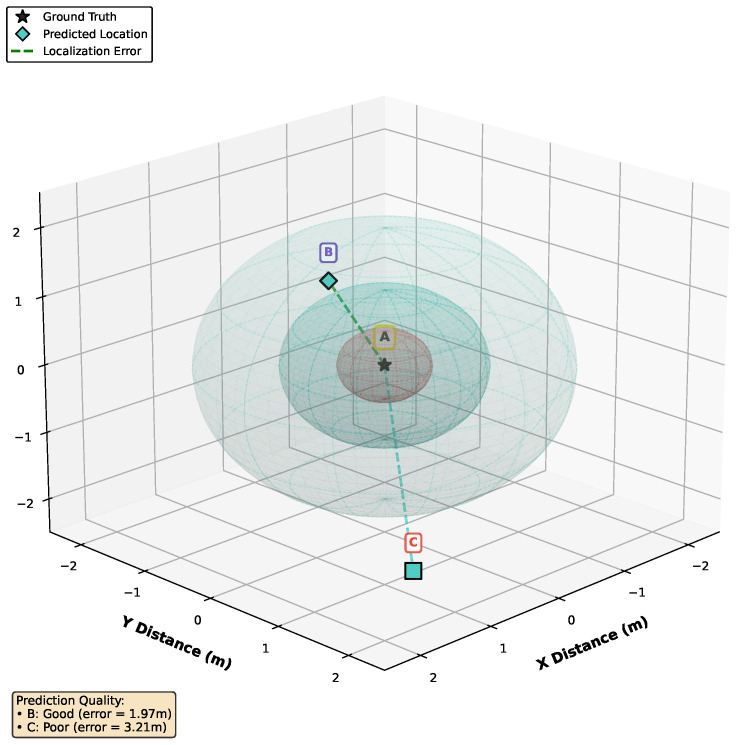
An example of multiple thresholds (radius values) used for localization. An object is considered detected correctly if it is within the spheres. In this figure, A is the ground truth, whereas B and C are predicted locations. The smaller the threshold, the more difficult an objected can be detected.

**Figure 6 sensors-26-00168-f006:**
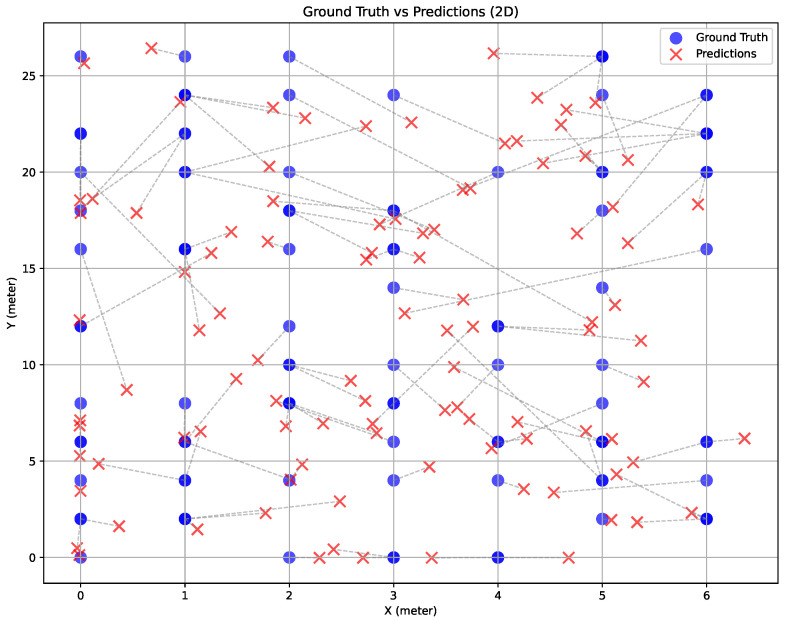
The plot of ground truth and predicted locations from SpaceNet in the Yeoju dataset. The dotted lines represent Euclidean distances. Feature: mel-spectrogram. Threshold: 2 m.

**Table 1 sensors-26-00168-t001:** Decomposition of layers in the SpaceNet model.

Module	Type	Input → Output	# Params
ResNet backbone	ResNet-18	4-ch → 512	11,179,648
Audio head FC1	Linear	512→256	131,328
Audio head FC2	Linear	256→128	32,896
Audio head FC3	Linear	128→29	3741
Spatial FC (dist)	Linear	4→4	20
Spatial FC (az)	Linear	4→4	20
Spatial FC (el)	Linear	4→4	20
Spatial aggregation	Linear + Tanh	4→1	5
Fusion FC1	Linear	32→16	528
Fusion FC2	Linear	16→8	136
Fusion FC3	Linear	8→3	27
Dropout	p=0.5	–	0
Learnable weight $w_{\mathrm{dist}}$	Scalar	1	1
Learnable weight $w_{\mathrm{az}}$	Scalar	1	1
Learnable weight $w_{\mathrm{el}}$	Scalar	1	1
Learnable weight $w_{\mathrm{spatial}}$	Scalar	1	1
Learnable weight $w_{\mathrm{audio}}$	Scalar	1	1

**Table 2 sensors-26-00168-t002:** Summary of Audio Datasets.

Dataset	Samples	Train Segments	Test Segments	Audio Format
Yeoju	14,708	51,478	7354	PCM, 44.1 kHz, mono, 16-bit
Yeongdong	1056	3324	900	PCM, 48 kHz, mono, 24-bit
Gyeonggi	3342	5468	7900	PCM, 48 kHz, mono, 24-bit

**Table 3 sensors-26-00168-t003:** Raw performance (no augmentation) comparison of SpaceNet and CHAWA models across datasets. Values indicate localization accuracy (%) at varying error thresholds. Bolded values represent the best performance for each dataset-threshold combination.

Dataset	Model	Feature	≤1.0	≤1.5	≤2.0
Yeongdong	SpaceNet	mel-spectra	9.3	**23.1**	**38.2**
	SpaceNet	mel-spectrogram	9.3	17.3	28.4
	CHAWA	mel-spectra	**9.8**	20.0	31.6
	CHAWA	mel-spectrogram	9.3	19.6	33.8
Yeoju	SpaceNet	mel-spectra	**53.9**	**65.2**	**73.1**
	SpaceNet	mel-spectrogram	32.8	50.3	64.8
	CHAWA	mel-spectra	5.5	11.4	18.7
	CHAWA	mel-spectrogram	4.4	10.0	16.1
Gyeonggi	SpaceNet	mel-spectra	28.0	49.4	66.5
	SpaceNet	mel-spectrogram	26.5	46.7	65.0
	CHAWA	mel-spectra	**28.3**	**50.8**	**67.5**
	CHAWA	mel-spectrogram	20.4	39.0	56.5

**Table 4 sensors-26-00168-t004:** Localization performance comparison between SpaceNet and CHAWA under rainy and non-rainy conditions.

Has Rain	SpaceNet		CHAWA	
	MAE (m)	RMSE (m)	MAE (m)	RMSE (m)
No	0.760	0.877	2.108	2.128
Yes	0.900	0.926	1.962	1.970

**Table 5 sensors-26-00168-t005:** Ablation study of SpaceNet on the Yeoju dataset for both 3D and 2D localization tasks under narrower error thresholds. The best performer at each threshold is highlighted in bold. The average R^2^ column shows the coefficient of determination for each model variant.

Dimension	Normalized	Spatial	Feature	≤1.0	≤1.5	≤2.0	Avg. R^2^
3D	✓	✓	mel-spectra	**53.9**	**65.2**	**73.1**	**0.53**
	✓	✗	mel-spectra	**53.9**	**65.2**	72.9	0.52
	✗	✗	mel-spectra	49.3	61.8	69.0	0.50
	✓	✓	mel-spectrogram	**32.8**	50.3	64.8	**0.57**
	✓	✗	mel-spectrogram	32.6	**51.4**	**65.7**	0.56
	✗	✗	mel-spectrogram	35.0	51.1	65.5	0.56
2D	✓	✓	mel-spectra	**52.8**	**66.3**	**74.2**	**0.81**
	✓	✗	mel-spectra	52.2	65.5	72.9	0.80
	✗	✗	mel-spectra	40.1	53.3	62.1	0.66
	✓	✓	mel-spectrogram	**37.2**	**56.9**	**71.1**	**0.86**
	✓	✗	mel-spectrogram	34.4	53.0	66.9	0.85
	✗	✗	mel-spectrogram	35.3	53.4	68.1	0.85

## Data Availability

The data presented in this study are available on request from the corresponding author.

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
