# Peer review of "SpaceNet: A Multimodal Fusion Architecture for Sound Source Localization in Disaster Response"

_sensors, 2025, doi:10.3390/s26010168_

Round 1

Reviewer 1 Report

Comments and Suggestions for Authors The article presents a method for Sound Source Localization that obtains good results in an environment with noise and reverberations. Unlike traditional methods that are based on signal processing techniques as computing the delays between the microphones recordings, this article uses a deep learning architecture that makes a fusion between audio features of microphones recordings and microphones locations.   Some observations about the presentation of the method and the results. It is presented that experiments were conducted in 2D and 3D settings. What does mean by 3D? Namely do the loudspeakers have a height above the ground? The same thing regarding microphones? The idea about the microphone presented on the page 10 is that they were positioned at the corners of each recording site. But nothing about the height. On the other hand, the elevation angle in 2D is zero. It is used for 2D implementation? What is the brand of the wireless microphones? How they transmit the recording data? What is the correspondence between the samples and segments in the Table 1? Because a neural network is used to both training and test the method, I think some details about it would be necessary. For instance: -the structure of the network-number and type of the layers, the optimization algorithm, number of epochs and iterations -the size of the input that mean the audio features (mel spectra and mel-spectrogram)   -results of the training process: the elapsed time, the training accuracy What is the response time? That means if the loudspeaker is moved with a small distance for instance 50 cm, how long it take such as the new coordinates to be available? Some observations about the document format:  On page 4, there is a reference to Fig. 4, but no reference was made to Fig. 2 and 3 until that point. Furthermore, Fig.4 is located two pages later. Similarly, Fig. 6 is on the page 11 but the reference to it is on the page 13. What does represent W and b within the equations on the page 6? What does represent the variable yi on the dataset (row 235). Maybe the speaker position? The significance of the blocks in the diagram of the fig. 1 is difficult to understand, the letters are too small. In fig. 6 the dimensions of x and y should be meters.

Author Response

We sincerely thank the reviewers for their constructive suggestions and insightful questions, which have helped us improve the quality and clarity of the manuscript.

Below are our responses to the questions from Reviewer #1.

Comments 1: It is presented that experiments were conducted in 2D and 3D settings. What does 3D mean? Namely do the loudspeakers have a height above the ground? The same thing regarding microphones? The idea about the microphone presented on the page 10 is that they were positioned at the corners of each recording site. But nothing about the height. On the other hand, the elevation angle in 2D is zero. It is used for 2D implementation?

Response: When we say “3D”, we mean that three dimensions (x, y, z) of the microphones and loudspeaker are utilized in model training (loss calculation) and inference (prediction of the speaker’s location). The loudspeakers do have heights above the ground (height > 0) and sometimes can be on the ground (height = 0). This also applies to the microphones.

Please have a look at the analysis of one dataset - Yeongdong as an example

Mic 1

Mic 2

Mic 3

Mic 4

Loudspeaker

Possible locations

(10.2, 1.3, 6.0)

and

(15.1, 7.5, 6.0)

(12.8, 3.8, 5.9)

and

(14.8, 6.1, 5.9)

and

(6, 1.7, 5.9)

and

(6.4, 1.1, 5.9)

(4.0, 2.0, 7.3)

(1.1, 5.6, 8.5)

and

(3.6, 3.0, 8.5)

(11.8, 2.2, 6.0)

and

(12.6, 3, 6.0)

and

(13.1, 2.7, 0.0)

and

(13.1, 3.7, 0.0)

(30 locations)

Please note that some microphones such as Mic 3 have one fixed location due to the difficulty of setting it in multiple locations in a high-risk environment.

Conversely, when we use “2D”, during model inference, we ignore the prediction of height and only consider (x, y) to measure the performance. Therefore, we did not actually ignore the elevation in the 2D benchmark, we still used this value for model training, we only ignored the z-dimension in the Euclidean distance calculation.

We have updated Sections Datasets and Experiments in the manuscript to clarify the definition and setting of 2D and 3D scenarios.

Comment 2: What is the brand of the wireless microphones? How they transmit the recording data? 

Response: The wireless microphones are RØDE Wireless GO II microphones. These microphones convert the analog sound from a recorded audio sample into a digital signal directly within the transmitter unit. This digital data is then encrypted and broadcast over the 2.4GHz frequency band, which is the same radio frequency range used by Wi-Fi.  Finally, the receiver picks up this encrypted data packet, decodes it back into audio, and sends it out to the Zoom R24 mixer.

We have updated the manuscript to describe the brands of the equipment used in the experiments, and explain how the wireless microphones transmit the recorded audio samples.  

Comment 3: What is the correspondence between the samples and segments in the Table 1? Because a neural network is used to both training and testing the method, I think some details about it would be necessary. For instance: 

-the structure of the network

-number and type of the layers, the optimization algorithm, number of epochs and iterations 

-the size of the input that mean the audio features (mel spectra and mel-spectrogram)   

-results of the training process: the elapsed time, the training accuracy 

Response: One audio sample comprises several segments. During the dataset preparation, we continuously played a long audio file containing different sounds, which may last up to 30 seconds. The audio samples are then broken into audio segments, for example, three seconds each for model training. This helps save time since we do not have to turn on and off the loudspeaker multiple times.

For the details of the training and testing, please refer to the followings

First, the structure of the network

  1. Backbone: ResNet-18–style CNN (audio feature extractor) The core is a ResNet with BasicBlocks, structurally equivalent to ResNet-18, with two adaptations: 4-channel input to consume the audio features, and classification head is replaced by a global pooling of a 512-D embedding. This backbone network converts multichannel audio input into a compact, high-level audio embedding. This embedding is gradually downsampled 512 → 256 → 128 → 29 to produce a compact embedding to be fused with the spatial information.
  2. Spatial feature branch: is a low-dimensional microphone geometry processing, which explicitly models distance, azimuth, and elevation.
  3. The fusion network: performs late fusion, which concatenates features of audio and spatial into 32-D embedding and continues to downsample 32 → 16 → 8 → 3. The Dropout (p = 0.5) is for regularization.

Second, the layers and their parameters

We use AdamW optimizer with the batch size of 128 and learning rate of 0.081. The number of epochs are 100 by default, with early stopping when there is no improvement of validation loss after 10 epochs. 

We briefly describe the layers as follows

- AdaptiveAvgPool2d: applied once to perform global average pooling; no trainable parameters.
- BatchNorm2d: used to normalize feature maps and stabilize training; 9,600 parameters.
- Conv2d: used for spatial feature extraction; 11,170,048 parameters.
- MaxPool2d: applied once to downsample feature maps
- Tanh: used once as a bounded activation function, this is particularly useful for localization of sound source since it is more balanced than ReLU.

For more details, please refer to the newly updated manuscript.

Third, the input size.

For three-second segments, melspectrogram has a size of 512 x 259, melspectra has a size of 4 x 128. During training, the input to the neural networks would have the shape of [B x C x W x H] where W x H can be either 512 x 259 or 4 x 128. B is the batch size, which is defaulted to 128 in our experiments. The channel C is always 1 since this task does not involve RGB images.

Fourth, for the training process. 

The parameters (mean and variance) for feature normalization are calculated once and are reusable across training and testing sessions. Depending on the size of the dataset, the calculation time can vary, for example given 5427 samples, it takes 72 seconds to calculate the mean and variance of these samples. This is the newest result we have just calculated on the Goyang’s training dataset.

While it is not easy to calculate the exact time elapsed between two epochs since the total training and testing time would include the “initialization”, we can deduce the time for a single epoch by deducting the execution time for two epochs by the execution time for one epoch.

For two epochs

For one epoch

Time elapsed between 2 epochs

Execution time

486.22 secs

267.95 secs

218.27 secs

User CPU time

19.61 mins

10.66 mins

8.95 mins

System CPU time

59.18 mins

31.89 mins 

27.29 mins

Please be informed that some of the details have already been mentioned in the previous manuscript, for example in Section 5.1 we mentioned the optimization algorithm. We understand that the details are not in one place making it hard to follow. Therefore, in this revision, we have also prepared a new table named “Decomposition of layers in SpaceNet model” in the manuscript to summarize such details.

Comments 4: What is the response time? That means if the loudspeaker is moved with a small distance for instance 50 cm, how long does it take such as the new coordinates to be available? 

Response: This is a meaningful question. While the proposed system was not inherently designed for dynamic tracking, we do expect the localization results to change as the loudspeaker is moved. The response time thus can be controlled to a certain degree. 

For instance, in streaming manner, the audio samples are “buffered” and get submitted to the system every 3 seconds (configurable). The processing and inference would take less than 1 second. If we take into account the delay of transmission and hardware (the mixer and the connected computer) then we expect 0.5 second more. 

To conclude, it would take approximately 4.5 seconds for the new coordinates to be available. If we deploy this on the cloud (remote server) then it may take additional 1~2 seconds per inference step.

Please note that the use case here is for localization of victims and we normally assume the locations are fixed. We however acknowledge the use of dynamic tracking of a moving object.

Comments 5: Some observations about the document format:  On page 4, there is a reference to Fig. 4, but no reference was made to Fig. 2 and 3 until that point. Furthermore, Fig.4 is located two pages later. Similarly, Fig. 6 is on the page 11 but the reference to it is on the page 13. 

Response: We acknowledge the issue with the relative positions of the Figures and their references.

This is due to the misplacement of LaTeX code that is used to render the Figures and text.

We have fixed it in this revision. Please be informed that due to the formatting of LaTeX toolings, sometimes we cannot put the same Figure and its reference on the same page, but we try to position them as close as possible. For the current state, most Figures and their references are only 1 page away (except for Figure 6).

Comments 6: What does represent W and b within the equations on the page 6? What does represent the variable yi on the dataset (row 235). Maybe the speaker position? 

Response: W and b in the equation on page 6 represent the weight and bias. These two variables are used in neural networks to define the learnable values - also known as hyperparameters connecting deep layers.

And you are correct that the variable “yi” is the speaker position. Just to be clear that normalization steps do not involve “yi”, its existence there is just for completeness of the definition of the dataset D.

We have clarified these variables in the revised manuscript.

Comment 7: The significance of the blocks in the diagram of the fig. 1 is difficult to understand, the letters are too small. In fig. 6 the dimensions of x and y should be meters.

Response: We have enlarged the letters in Fig. 1, the workflow is that there are inputs including multi-channel audio samples (recorded by the microphones) and the positions of those microphones. Meanwhile the speaker positions are the labels which are only available in training. The “dual processing streams” block is the network that processes the inputs. 

Regarding the units in Fig. 6, we have added “meters” to the axes of X and Y.

Reviewer 2 Report

Comments and Suggestions for Authors

The authors introduced SpaceNet, a multimodal neural network whose main goal is to the localization of sound sources where spatial characteristics of the environment are incorporated.

The work introduced by the authors builds on and even compares the performance of their proposal with previous works. Multimodal is a relevant topic and the authors have made a serious work, in particular, the multimodal normalization adds stability to the operation fo their proposal. 

The work also compises results from various scenarios. However, it will be worth to explain the limitations of the results and the impact that the limited number of microphones may have on the results.  Will it be possible from the results to identify not only the number of microphones and the best location. At least for the environment under consideration. But since the ultimate goal is to be used in disaster situation, is it possible to provide some guidelines on what to do?

As a final point, the authors may consider to change the name of their proposal  SpaceNet has been used by other authors, using exactly the same numenclature. See for instance:

SpaceNet: Make Free Space For Continual Learning by Ghada Sokara,  Decebal Constantin Mocanua and Mykola Pechenizkiya, Journal of Neurocomputing.

Author Response

We sincerely thank the reviewers for their constructive suggestions and insightful questions, which have helped us improve the quality and clarity of the manuscript.

Below are our responses to the questions from Reviewer #2.

Comments 1: The work also comprises results from various scenarios. However, it will be worth to explain the limitations of the results and the impact that the limited number of microphones may have on the results.  Will it be possible from the results to identify not only the number of microphones and the best location. At least for the environment under consideration. But since the ultimate goal is to be used in disaster situation, is it possible to provide some guidelines on what to do?

Response:

On Mic Number: We used a fixed set (e.g., 4 sensors). While we did not explicitly test the higher number of microphones due to the constraint of purchased products (RØDE Wireless GO II and Zoom R24 mixer. We have two receivers, each receiver corresponds to two microphones (4 microphones in total) . Our spatial branch is designed to scale with the number of microphones. A limitation is that we haven't defined the 'minimum' number required for robustness.

On the best location: As mentioned in the manuscript, we found that 3D performance suffered when Z-axis variance was low. Therefore, a common strategy is that sensors must be placed at varying heights to ensure accurate 3D localization. 

On guidelines for disaster situations: For the deployment of microphones at the disaster site, since SpaceNet encodes geometry explicitly, it doesn't require a perfect grid. The guideline is to deploy sensors in an ad-hoc, distributed manner, but maximize the spatial spread to cover the search area. According to our experiments, the ideal area should not be larger than 20m x 20m and no deeper than 6m.

For the limitations of the microphone settings, we have added them to the end of Section Experiments (Section 5).

Comments 2: As a final point, the authors may consider to change the name of their proposal  SpaceNet has been used by other authors, using exactly the same nomenclature. See for instance:

SpaceNet: Make Free Space For Continual Learning by Ghada Sokara,  Decebal Constantin Mocanua and Mykola Pechenizkiya, Journal of Neurocomputing.

Response: Firstly we really appreciate the suggestion, in fact after our submission, we were aware of several studies using “SpaceNet” as their nomenclature (including the one mentioned above).

For instances, we found the following studies that also adopted the name “SpaceNet”

  1. Lee, G., Jordan, E., Shishko, R., de Weck, O., Armar, N., & Siddiqi, A. (2008, September). SpaceNet: modeling and simulating space logistics. In AIAA SPACE 2008 conference & exposition (p. 7747).
  2. Van Etten, Adam, Dave Lindenbaum, and Todd M. Bacastow. "Spacenet: A remote sensing dataset and challenge series." arXiv preprint arXiv:1807.01232 (2018).
  3. Sokar, G., Mocanu, D. C., & Pechenizkiy, M. (2021). Spacenet: Make free space for continual learning. Neurocomputing, 439, 1-11.

For our study, SpaceNet implies Spatial information and a submodule of residual network (ResNet18), which conveys its own meaning. The purpose of naming is purely for the convenience of the readers of the menuscript. We did not intentionally choose the name to cause confusion, neither did we plan to patent such a name. 

We appreciate the reviewer’s suggestion and will consider the new name for SpaceNet during our camera-ready version. 

In this revision, we have modified the manuscript to explain about the nomenclature in the Section of SpaceNet to clarify the possible confusion.

Reviewer 3 Report

Comments and Suggestions for Authors

This article is both interesting and relevant. Its main contribution is the development and experimental testing of a new, specially adapted multimodal SpaceNet architecture for localising sound sources in complex natural disaster conditions. The authors proposed a model combining audio features and sensor geometry via a two-branch structure in which the spatial channel explicitly encodes microphone distances, azimuths and angles. They demonstrated that using ILD signals as primary features provides greater reliability in reverberant and asynchronous environments than complex temporal features do. They showed that SpaceNet is more accurate than the baseline CHAWA model and models trained on full mel-spectrograms. At the same time, the computational cost is reduced by a factor of 24. Another significant contribution is a novel normalization scheme that stabilises multimodal learning and effectively combines spatial and audio features.

However, the article lacks detailed information. Some comments on the article:

1) The inscriptions of the structural diagram blocks in Figure 1 are disproportionately small relative to the size of the figure. Therefore, they are difficult to read as the letters are very small. Please correct this.

2) The authors conducted an experiment. However, there is no information on the accuracy with which the sound source's location can be determined. They should provide numerical values for this accuracy.

3) All experiments were conducted on arrays with four microphones located at the corners. Is the model proposed by the authors only correct for this spatial arrangement of microphones? Can this model be applied to other microphone array configurations?

4) In the event of a natural disaster, the acoustic background can be extremely noisy. As the authors wrote on lines 52–55, significant background noise can seriously distort spatial signals. In their description of the dataset, the authors state that all datasets include realistic environmental noise artefacts (lines 269–270). However, the article presents no numerical results from the analysis or testing of the effect of noise on the accuracy of signal-source localisation. What signal-to-noise ratio did the authors use for the experiment?

5) The authors of the article claimed that SpaceNet 'surpasses' the previous model, but did not provide any specific figures or statistical indicators to support this claim. It would be helpful if the authors could provide quantitative indicators of the neural network's effectiveness, such as mean squared error (MSE) and root mean squared error (RMSE), or other standard metrics. The authors claim that SpaceNet is a multimodal deep neural network architecture that outperforms the baseline model. Numerical metrics would clearly confirm this advantage.

6) Did the authors control the environmental conditions (temperature, humidity, wind direction and strength) during the experiment? How would changes in these environmental parameters affect the accuracy of sound localisation using the method proposed by the authors?

Author Response

We sincerely thank the reviewers for their constructive suggestions and insightful questions, which have helped us improve the quality and clarity of the manuscript.

Below are our responses to the questions from Reviewer #3.

Comments 1: The inscriptions of the structural diagram blocks in Figure 1 are disproportionately small relative to the size of the figure. Therefore, they are difficult to read as the letters are very small. Please correct this.

Response: We have enlarged the inscriptions of Figure 1.

Comments 2: The authors conducted an experiment. However, there is no information on the accuracy with which the sound source's location can be determined. They should provide numerical values for this accuracy.

Response: Besides the “localization accuracy” and “coefficient of determination” mentioned in the Experiments section of the original manuscript, we have added the “localization error” (MAE and RMSE) - which determines how far or close the prediction location is to the speaker’s location. Please refer to the next responses for more details (at Comments 5).

Comments 3: All experiments were conducted on arrays with four microphones located at the corners. Is the model proposed by the authors only correct for this spatial arrangement of microphones? Can this model be applied to other microphone array configurations?

Response: For the current settings, we deploy the microphones in an ad-hoc, distributed manner, but maximize the spatial spread to cover the search area. Therefore  oftentimes these microphones are located in the corners.

The reason we used four microphones is because of the “cone of confusion”, which  is a phenomenon in both human hearing and microphone array processing where multiple sound source locations produce nearly identical interaural time differences (ITDs) and interaural level differences (ILDs). Given the environment with noises, we did not expect the number of microphones fewer than four would work. This has been explored in the paper of wav2pos - the work is open source and we did have some discussions with the authors regarding the masked positions.

Our study can work and may produce even better results if there are more microphones added, this is theoretically possible and can also be confirmed by related work such as wav2pos with their simulation using pyroomacoustics framework. To answer your question, yes, this model can absolutely be applied to other microphone array configurations. The other reason that restricted us from adding more microphones  was the cost of the devices, which can be abundant.

Comments 4: In the event of a natural disaster, the acoustic background can be extremely noisy. As the authors wrote on lines 52–55, significant background noise can seriously distort spatial signals. In their description of the dataset, the authors state that all datasets include realistic environmental noise artefacts (lines 269–270). However, the article presents no numerical results from the analysis or testing of the effect of noise on the accuracy of signal-source localisation. What signal-to-noise ratio did the authors use for the experiment?

Response: Regarding the noise, we measure the signal-to-noise ratio above 0 dB most of the time. The manuscript is a result from a research project monitored by a third-party designated from the government, and our aim is to achieve 80% of localization accuracy given 0 dB signal-to-noise ratio.

We however did not control other factors such as weather temperature, humidity, wind direction, rain, nearby construction or random operations (for example, helicopters sometimes flew above during our recording sessions). We only had some basic measurements, for example the temperatures range between 0~10 Celsius degrees in the early winter and between 22~27 Celsius degrees during the spring to the summer. The humidity is relatively low between 30% to 50% at day time, which can get lower in the winter. Most of the time, it was not raining. However the data reported in Comment 2 were measured in a light rain.

Comments 5: The authors of the article claimed that SpaceNet 'surpasses' the previous model, but did not provide any specific figures or statistical indicators to support this claim. It would be helpful if the authors could provide quantitative indicators of the neural network's effectiveness, such as mean squared error (MSE) and root mean squared error (RMSE), or other standard metrics. The authors claim that SpaceNet is a multimodal deep neural network architecture that outperforms the baseline model. Numerical metrics would clearly confirm this advantage.

Response: Below is the validation loss during training CHAWA and SpaceNet, SpaceNet has a more stable training session given the same configuration with CHAWA. For instance, the loss function is huber_loss, the optimizer is adamw, the number of epoch is 100, the batch size is 128 and the learning rate is 0.00081.

As a result, SpaceNet (Green color) was able to continue with the optimization, while CHAWA (Orange color) paused the training at epoch 80 due to “early stopping” (validation loss did not improve after 10 epochs).

SpaceNet

CHAWA

Has Rain

MAE (meter)

RMSE (meter)

MAE (meter)

RMSE (meter)

no

0.760

0.877

2.108

2.128

yes

0.90

0.926

1.962

1.970

Also, for the same epochs, the two models have the following coefficient of determination across the 3 axes (higher is better). 

CHAWA R2 score: [0.5119997  0.56711805 0.7576071 ], average R2 = 0.612

SpaceNet R2 score: [0.4688704  0.83893037 0.9044233 ], average R2 = 0.737

On average, SpaceNet has a higher coefficient of determination compared to CHAWA, which implies that in SpaceNet, the predicted locations are closer to the ground truths than that of CHAWA.

Comments 6: Did the authors control the environmental conditions (temperature, humidity, wind direction and strength) during the experiment? How would changes in these environmental parameters affect the accuracy of sound localisation using the method proposed by the authors?

Response: Besides our answers in “Comments 4”, we would like to emphasize that we did not control the environment conditions. We however performed some audio recordings more than one time on different dates at the same location. Please refer to the followings

At the same disaster site, the conditions between  Day 1 and Day 2 were almost the same except for one important factor: Day 2 had rain. The raindrop sounds obviously affected the quality of localization. Therefore, the MAE and RMSE of Day 1 are lower than Day 2’s.

MAE (meter)

RMSE (meter)

1-meter Loc. Acc (%) 

Rain

Day 1

0.760

0.877

75%

No

Day 2

0.90

0.926

60%

Yes

We did not have the capacity to measure other factors such as the strength and direction of the winds, however we do believe that noise caused by raindrops would have more impact on this experiment.

For completeness, we have added this discussion in the manuscript.